# Bioactive Efficacy of Novel Carboxylic Acid from Halophilic *Pseudomonas aeruginosa* against Methicillin-Resistant *Staphylococcus aureus*

**DOI:** 10.3390/metabo12111094

**Published:** 2022-11-10

**Authors:** Henciya Santhaseelan, Vengateshwaran Thasu Dinakaran, Balasubramaniyan Sakthivel, Maharaja Somasundaram, Kaviarasan Thanamegam, Velmurugan Devendiran, Hans-Uwe Dahms, Arthur James Rathinam

**Affiliations:** 1Department of Marine Science, Bharathidasan University, Tiruchirappalli 620024, India; 2Drug Discovery and Development Research Group, Department of Pharmaceutical Technology, University College of Engineering Bharathidasan Institute of Technology Campus, Anna University, Tiruchirappalli 620024, India; 3Department of Biomedical Science, Bharathidasan University, Tiruchirappalli 620024, India; 4Department of Biomedical Science and Environmental Biology, Kaohsiung Medical University, Kaohsiung 80708, Taiwan; 5Research Center for Precision Environmental Medicine, Kaohsiung Medical University, Kaohsiung 80708, Taiwan; 6Department of Marine Biotechnology and Resources, National Sun-Yat-sen University, Kaohsiung 80424, Taiwan

**Keywords:** antibiotic resistance, MIC, antimicrobial, antioxidant, gene expression, in silico, binding affinity, toxicity

## Abstract

Methicillin-resistant *Staphylococcus aureus* (MRSA) infections are increasingly causing morbidity and mortality; thus, drugs with multifunctional efficacy against MRSA are needed. We extracted a novel compound from the halophilic *Pseudomonas aeruginosa* using an ethyl acetate (HPAEtOAcE). followed by purification and structure elucidation through HPLC, LCMS, and ^1^H and ^13^C NMR, revealing the novel 5-(1*H*-indol-3-yl)-4-pentyl-1,3-oxazole-2-carboxylic acid (Compound **1**). Molecular docking of the compound against the MRSA PS (pantothenate synthetase) protein was confirmed using the CDOCKER algorithm in BDS software with specific binding to the amino acids Arg (B:188) and Lys (B:150) through covalent hydrogen bonding. Molecular dynamic simulation of RMSD revealed that the compound–protein complex was stabilized. The proficient bioactivities against MRSA were attained by the HPAEtOAcE, including MIC and MBCs, which were 0.64 and 1.24 µg/mL, respectively; 100% biomass inhibition and 99.84% biofilm inhibition were observed with decayed effects by CLSM and SEM at 48 h. The *hla*, *IrgA*, and *SpA* MRSA genes were downregulated in RT-PCR. Non-hemolytic and antioxidant potential in the DPPH assay were observed at 10 mg/mL and IC_50_ 29.75 ± 0.38 by the HPAEtOAcE. In vitro growth inhibition assays on MRSA were strongly supported by in silico molecular docking; Lipinski’s rule on drug-likeness and ADMET toxicity prediction indicated the nontoxic nature of compound.

## 1. Introduction

The human clinical sector has consistently faced issues on antibiotic resistance and the more prevalent methicillin-resistant *Staphylococcus aureus* (MRSA) [1,2]. It can cause food poisoning to fatal necrotizing pneumoniae and can increase load of high-risk assessments [3,4]. *Staphylococcus aureus* is well-adapted to its human host and the health-care environment [5]; it is a leading cause of endocarditis, bacteremia, osteomyelitis, and skin and soft tissue infections. *S. aureus* quickly became well-known as a leading source of health-care-related infections [6]; although it is capable of communally surviving in the environment, a favorable habitat causes the bacterium to become even more infectious and spread across multiple organ systems. MRSA is prevalent in the majority of Asian hospitals, and several Asian countries have some of the highest MRSA prevalence rates in the world [7]. According to the CDC, the pandemic of COVID-19 in the United States has considerably increased MRSA infections by 13% in 2020 compared with 2019 [8]. MRSA infections had higher mortality rates than AIDS and Parkinson’s disease, as demarcated by the American Infectious Disease Society [9]. The WHO states that people with MRSA infections are more likely to die than people with drug-sensitive infections, and MRSA is the dominant species causing blood stream infections coupled with 3rd generation cephalosporin-resistant *E. coli* [10]. Several reports show that MRSA plays two categorized roles: hospital-acquired infection (HAI) and community-acquired infection (CAI) [11]. MRSA can cause nosocomial infections in HAI, which typically require surgical or insidious therapeutic care during hospital stays that rarely causes infections in healthy (non-hospitalized) individuals [12]. CAI mainly causes skin and soft tissue infections that may occur in the common population in unhygienic environments [13]; also, colonization by MRSA leads to increasing infections [14]. Chronic infections, combined with biofilm formation, can persist in host tissue and implanted materials, including bone, catheters, base makers, and prosthetic joints causing osteomyelitis illness, heart valve endocarditis, etc. [15]. Implanted materials turn out to be fleeced with MRSA host protein and matrix-binding proteins over the surface of *S. aureus*. This increases the attachment toward these proteins and develops biofilm formation in the case of infections with materials; device removal is the only way to treat the infection [16]. Several factors have been involved in biofilm formation, including bacterial density, stress responses, physiological properties, antibiotic resistance, EPS (exopolysaccharides) neutralizing the antibiotics, enzyme production, and QS (quorum sensing) ability [17]. However, the virulence of MRSA contains various characteristics, including adhesins, immunomodulators, toxins, and enzymes, as well as the high-risk factor PVL (Panton–Valentine leukocidin), which is a toxin associated with severe necrotizing pneumonia [18,19]. PVL is encoded by two genes, *luk-S-PV* and *luk-F-PV*, which also provides cytotoxicity to mammalian cells [20]. In addition, serine protease ETs (exfoliative toxins) play a role in host colonization via recognition of desmosome proteins in the skin and cause injured mucosa, also resulting in local epidermal infections and generalized diseases; serotypes such as ETA, ETB, and ETD increase the severity of infection [21]. Apart from these virulent factors, the plasmids, i.e., the cassette chromosome of *S. aureus* (SCC*mec*) and mecA ABR gene-coding proteins are generally impeded by β-lactam antibiotics via inactivating transpeptidases, which are important for cell wall synthesis [22]. The multidimensional virulence profile of MRSA presents many serious threats, including toxic shock syndrome, which is a life-threatening illness after various surgeries [23]. These threats harm human populations and show the urgent need to discover and propagate various bioactive compounds from natural sources to control/diminish the virulence of MRSA to protect at-risk groups from these harmful microbes. Currently, extremophilic microbes are receiving growing attention for being a source of numerous novel biotechnological products with antimicrobial, antioxidant, and antitumor effects, as well as food technology and industrial products that show cold adaptiveness, thermostability, salt tolerance, and enzyme productions [24,25,26,27]. Halophilic microbes have been noted for their outstanding bioactivity against various drug-resistant pathogens that also possess several bioactivities [28,29,30], such as ethyl acetate extracts originating from halophilic *Pseudomonas aeruginosa* that endow better antibacterial activity against drug-resistant microbes, such as *Klebsiella quasivariicola* and *E. coli* isolated from foot infections of patients of diabetes [31]. *P. aeruginosa* is an opportunistic pathogen and causes frequent infections in clinical settings; [32] interestingly, the halophilic habitat of this organism has been shown to survive in high salt concentrations for a long time, with several genes showing adaptation for saline environments [33]. Remarkable bioactive compounds have been developed from the *Pseudomonas* genus, including quinoline, phenanthrene, pyrroles, phenazine, phloroglucinol, pseudopeptide pyrrolidinedione, bushrin, and zafrin, and bioactive organo copper has also been extracted from *P. aeruginosa*, which can inhibit the plant pathogen *Xanthomonas citri* subsp. *citri* [34,35]. Hence, the current study investigates the identification of a particular bioactive compound present in the ethyl acetate extract of halophilic *P. aeruginosa*, which was responsible for previously reported antibacterial properties [31]. We studied this compound using chromatographic techniques and its distinct antibacterial role was evaluated against MRSA via various in vitro and in silico approaches, along with ADMET toxicity prediction.

## 2. Materials and Methods

### 2.1. Methicillin-Resistant Staphylococcus Aureus

The methicillin-resistant *S. aureus* (MRSA) was collected from KAP Viswanatham Government Medical College, Tiruchirappalli-620001, and its resistance was confirmed using a methicillin strip (HiMedia-MD031 MET (B)). The MRSA strain was cultured in a brain heart infusion (BHI) (Hi-media, Pvt Ltd., Mumbai, India) medium and stored in a glycerol stock (30%) at −20 °C for future analysis. 

### 2.2. Saltpan Bacteria

Bacterial isolation from saltpans and *Pseudomonas aeruginosa* isolation and bioactive extract preparation was previously reported by Henciya et al. [31]. Briefly, the supernatant of *P. aeruginosa* was derived by centrifuge at 6000 rpm for 10 min; an equal volume of EtOAc was added into the separating funnel containing the supernatant. After a fine shaking every 1 h, the colorless phase was taken and condensed using a rotary evaporator (RV3V-C-IKA). The final product was analyzed through gas chromatography-mass spectrometry, which revealed the chemical constitution, including hexadecanoic acid, ethyl ester, N-hexadecanoic acid, octadecanoic acid, ethyl ester diethyl phthalate, and some other acid derivatives [31]. The active EtOAc extract derived from the halophilic *P. aeruginosa* was used in the current study, for further screening of in vitro antibacterial activity.

### 2.3. Preliminary Antibacterial Activity 

The MRSA strain was cultivated in Muller Hinton Broth (MHB) sustained at 0.5 McFarland turbidity standards (10^8^ cfu/mL con). The cultivated strain was spread over the surface of the MHA plate. The wells were shaped over the MHA surface with a 6 mm diameter, and crude EtOAc (ethyl acetate) extract was added to the wells at different concentrations, including 20, 40, 60, 80, 100, 120, and 140 µL, respectively. The plates were then incubated at 37 °C for 24 h, and inhibition zones were measured using the Hi Antibiotic zone scale-^TM^C (Himedia Laboratories Pvt Limited, Mumbai, India).

### 2.4. Minimum Inhibitory Concentration (MIC) and Minimum Bactericidal Concentration (MBC) Determination 

The MIC and MBC of EtOAc crude extract were determined by using a micro broth dilution technique with 96-well plates in duplicate, followed by CLSI (Clinical and Laboratory Standards Institute) guidelines with a modified protocol of Buzgaia et al. [36]. Before that, for homogeneous distribution of the extract, the mixture was vortexed using a cyclomixer (REMI Pvt Ltd., Thane, India). Briefly, MHB (Muller Hinton Broth) medium-added wells were maintained as controls, including 100 µL MHB as a sterility control; MHB + 100 µL culture inocula (10^6^ CFU/mL) as a negative control; and MHB with the antibiotic clindamycin as a positive control (1st–3rd well). An aliquot of 100 µL from the fraction of 1000 µg/mL EtOAc extract was added to the 12th well, and two-fold dilutions were made from the 12th well down to the 4th well (2.5–0.01 µg/mL); therefore, the 12th well contained the highest concentration of the extract at 2.5 mg/mL. Each well was filled with 100 µL of an overnight culture of MRSA (10^6^ CFU/mL). The plates were then incubated at 37 °C for 24 h and the MIC was determined as the lowest concentration of the EtOAc extract that completely inhibited bacterial growth. A further 10 µL aliquot from each well was subcultured over the MHA plates, and the plates were incubated for 24 h at 37 °C to determine the MBC as the lowest concentration that completely killed the bacterial strains. 

### 2.5. Microbial Biomass Inhibition Assay

The assay was performed by mixing the active extract and monitoring cell growth [37]. Briefly, 150 and 75 µg/mL of EtOAc crude extract were separately mixed into 50 mL conical flasks with 5 mL of nutrient broth containing inoculated MRSA (10^6^ CFU/mL), and the flasks were incubated at 37 °C for 16–18 h on a shaking incubator. A flask without extract was maintained as a positive control. After incubation, the culture was centrifuged at 10,000 rpm for 10 min; after washing, the pellet was re-suspended in distilled water to measure the OD at 660 nm. Variations between the positive control and treated culture were used to calculate the percentage of inhibition on cell suspension biomass:% of inhibition = (control cells − treated/control) ∗ 100 

### 2.6. Bacterial Cell Live and Dead Assay 

The assay was performed using the protocol provided by the Live/Dead Bac Light Bacterial viability kit, (Thermo Fisher Scientific). Briefly, the later log phase bacterial cultures (control and treated) were taken from the biomass inhibition assay (75 and 150 µg/mL fixed) and centrifuged at 10,000× *g* × 10 min. The supernatant was removed and the pellet was resuspended with 2 mL wash buffer and slight shaking was performed every 15 min. The tubes were centrifuged again at 10,000× *g* × 10 min and washed twice using a washing buffer. Equal volumes of SYTO9 and PI (propidium iodide) were prepared in a micro-centrifuge tube according to the manufacturer’s instructions, and 3 µL of this mixed solution was added to 1 mL of bacterial suspensions and incubated in the dark at room temperature for 15 min. An amount of 5 µL of the sample dye mixture was placed on a microscopic slide and fluorescence intensity was assessed at the cellular level using epifluorescent microscopy (Optika, XDS-2+M-795, Italy), at 480 for SYTO9 and 490 for PI laser line image with 60X Objective; the presence of live/dead cells were observed. 

### 2.7. Biofilm Inhibition Assay 

The anti-biofilm activity of the EtOAc crude extract was performed, according to Rizzo et al. [38], using 6-well plates with experimental setups in duplicate. Briefly, 1 mL of TSB (Trypticase Soy Broth) medium was added to the wells and 100 µL of the MRSA strain was inoculated before the bacterial OD was adjusted to 0.1 at 600 nm. The control wells were maintained, including the medium alone as a sterility control; the medium and bacterial inoculum as a negative control; the medium and inoculum with clindamycin antibiotics as a positive control. The 150 µg/mL of HPA EtOAc crude extract was added into the wells based on the microbial biomass inhibition assay. Furthermore, the plates were incubated at 37 °C for 24–48 h to allow the cultures to form biofilms at the bottom and kept for incubation until 48 h. At intervals of 6, 12, 24, and 48 h, the appropriate OD was taken at 600 nm using the following procedure. After incubation, the media were carefully removed, along with planktonic free-floating cells from the plates, using a micro-tip, and non-adherent cells were removed by gentle washing with sterile water. The remaining attached cells were prepared as a suspension using 0.3% PBS solution and the OD was taken using a UV-Vis spectrophotometer (Shimadzu, UV-1900, Kyoto, Japan).

### 2.8. Confocal Laser Scanning Microscopy (CLSM) 

Similar experimental setups were prepared according to a biofilm inhibition assay, followed by a modified protocol of Shinde et al. [39]. Briefly, coverslips were placed into the six-well plates and MRSA biofilm was allowed to grow, as detailed above in the biofilm inhibition assay, and the wells were washed using 0.75% NaCl solution followed by gentle tapping with tissue paper to maximize drainage. The SYTO9 dye was used to stain the grown cells and the staining was made from a stock of 30 mM dye. Dye solution (20 µL) was added into the wells and kept for 15 min incubation. The plates were washed with Milli Q water and carefully drained to remove excess water. The coverslips from the wells were carefully removed and placed on the slide, followed by applying mounting oil (BacLight). The formation of biofilms at fluorescence was visualized under CLSM (Carl Zeiss, model LSM710, Jena Germany, (helium-neon laser)) at 625 nm, and the images were noted for the confirmation study.

### 2.9. Scanning Electron Microscopy 

The mid-exponential growth phase of the control and 75 and 150 µg/mL (microbial biomass inhibition assay fixed) treated MRSA was diluted with salt-free LB medium. The cells were centrifuged at 6000 rpm for 10 min and the pellet was prefixed with 2% glutaraldehyde, followed by washing with distilled water and PBS. The samples were washed in a graded ethanol series (30–100%) and then air-dried. Washing and fixing of reagents were done with 0.15 M sodium phosphate buffer (pH 7.2). A sample smear was prepared over the SEM glass slide and a pinch of platinum was sputtered over the samples to avoid charging by the instrument. Microscopic model EVO-18 (Carl Zeiss, Jena, Germany) was used and secondary electron images were taken at low electron energies between 2 and 2.5 keV.

### 2.10. Hemolytic Activity

Potential EtOAc crude extracts were screened for anti-hemolytic activity by spectrophotometric analysis. Briefly, 5 mL of a human blood sample was centrifuged at 1500 rpm for 5 min after collecting it into the EDTA vials. The supernatant was removed, and the pellet was washed thrice using 0.2 M PBS (pH 7.4) and re-suspended with 0.5% saline solution. The EtOAc crude extracts at various concentrations, including 1, 5, 10, and 20 mg were added to the 1 mL erythrocyte suspension and incubated for 1 h at RT. The buffer solution made with 250µL H_2_O_2_ was combined with the erythrocyte suspension was a positive control. The PBS with cell suspension was maintained as a negative control. Furthermore, the samples were centrifuged at 1500 rpm for 10 min and absorbance was taken at OD_540_ nm by spectrophotometer. Each experimental setup was performed in triplicate and various fractions providing inhibitory responses were calculated using the formula given below to elevate the inhibition percentage.
% of Hemolysis = (Abs. of Positive Control) − (Abs. of Test Sample)/(Abs. of Positive Control) × 100 

### 2.11. Antioxidant Activity DPPH (2,2-Diphenyl-1-Picrylhydrazyl) Assay

A free radical scavenging method was used to measure the antioxidant activity of the EtOAc crude extract using DPPH [40]; different concentrations of the HPAEtOAcE crude extract (5 mg/5 mL stock concentration; 10–60 µL) were prepared and dissolved in methanol, along with 100 µL of DPPH (3 mg/60 mL), and was added to a 96-well microtiter plate. The reaction was performed in triplicate and the plate was kept in the dark for 30 min at RT. Ascorbic acid was used as a positive control by maintaining a similar experimental setup and DPPH was used as a blank. The color intensity was observed at OD_517_ nm, applied by an Elisa microplate reader (Biotek, Winooski, VT, USA), and the inhibition percentage was calculated using the following formula:% of Inhibition = (Absorbance of control − Absorbance of sample)/(Absorbance of control) × 100 

The IC_50_ value was calculated using the logarithmic value of concentrations and pertained to the nonlinear regression equation [41] through the log (inhibitor) vs. normalized response-variable slope equation.

### 2.12. RT (Real Time)—PCR 

MRSA strains were grown in TSB containing MIC of the EtOAc crude extract, and the control was maintained as a strain without extract. Total RNA was extracted from cultures after 24 h, and cDNA synthesis was carried out using a QIAGEN RNA-PCR kit (West Caldwell, NJ07006) in RT-PCR (Applied Biosystems StepOne Plus™) with SYBER Green detection. The reaction mixtures of 20 µL were performed using qPCR QIAGEN Master Mix. The assays were carried out in triplicate and the primers used for the experimental cycle were given in Table 1.

### 2.13. Purification and Structure Elucidation of Bioactive Compound

Analytical grade reagents were used in compound purification analysis. The EtOAc crude extract (10 g) was subjected to column chromatography (7734 silica gel, Merck, 60–120 mesh size) using hexane:EtOAc (3:2) solvent systems. The eluted fraction was collected and the single compound fraction was determined by TLC (thin-layer chromatography-silica gel-coated 60 matrix; L × W: 5 × 20 cm)) with an Rf value of 0.65. The TLC active spot was analyzed by HPLC ((RP-HPLC—Shimadzu LC-10 system, Kyoto, Japan) to determine the purity of content by dissolving it in methanol (1 Ml) and passing them through a 0.2 µM milli pore micro filter and methanol (80%):water:acetic acid (0.2%) was used as the mobile phase (RP-HPLC—Shimadzu LC-10 system, Kyoto, Japan). The C18 column was used for separation (100 × 4.60 mm, 2.6 μm, 100 Å), and the column temperature was maintained at 35 °C. The molecular weight of the purified compound was confirmed through LC-MS (liquid chromatography-mass spectrometry) ((LC-MS/MS—8045 Shimadzu, Japan) analysis, and chromatographic separation of the compound was obtained using the column XR-ODS III (1502.0 mm), with temperature and injection volume of 40 °C and 10 µL, respectively. The water containing 0.1% formic acid was used as the mobile phase with a flow rate of 0.40 mL/min. Further, ^1^H ^13^C NMR spectroscopy (Bruker BioSpin, Rheinstetten, Germany) was performed. The ^1^H NMR was operated at 400 MHz and ^13^C NMR was operated at 100 MHz, using DMSO as a solvent, and the spectra were recorded.

### 2.14. Molecular Docking of Pantothenate Synthetase Active Site 

Molecular docking for identifying the binding affinity of the purified compound was performed using the CDOCKER algorithm in the BIOVIA discovery studio 2017 (BDS) software. The crystallographic structure of the pantothenate synthetase (PS) protein of MRSA (PDB ID: 2 × 3F) was obtained from the protein data bank (http://www.rcsb.org/), and the water molecules were removed. Protein and isolated compound energy minimization were done by a smart minimizer algorithm, followed by the minimized protein and ligands that were submitted to CDOCKER for docking analysis. 

### 2.15. Molecular Dynamic Simulation

To confirm protein stability, RMSD and RMSF of the ligand–receptor complex in molecular docking were analyzed by the Standard Dynamics Cascade protocol in BDS. The protein–ligand complex obtained from the docking was used for the molecular simulation study by applying the charm force field with the solvation method. Standard parameters were programmed for MD simulations; heating was applied to the solvation system from 50 to 350 K as a target temperature and the equilibrium steps were followed. Analysis trajectory methodology was used to look at the angles of the residues. The 100 ns simulation was run, during which 1000 frames were saved to the trajectory. To determine the stability of the docked complex, the total energies of each confirmation in each time frame of the protein and docked protein were analyzed and compared.

### 2.16. ADMET and Toxicity Prediction

The predicted pharmacokinetic and pharmacodynamics, toxicity, and drug-likeness properties of the isolated compound were evaluated by ADMET and TOPKAT algorithm in BDS software. These algorithms have been mainly used to predict the drug-like properties and toxic fragments present in the isolated compound using FDA protocol.

### 2.17. Statistical Analysis

All experimental analysis including MIC, MBC, Microbial Biofilm, biomass inhibition, and anti-hemolytic analysis were performed in triplicate by statistical reference mean and standard deviation, and the antioxidant assay data were calculated using the SPSS 25 statistical package and analyzed using ANOVA, followed by Duncan’s multiple range test (level of significance *p* < 0.05).

## 3. Results

### 3.1. Antibacterial Activity, MIC and MBC

At all concentrations, no zone formation was observed in the methicillin antibiotic strip, confirming that hospital strains have methicillin resistance (Appendix A). The preliminary antibacterial activity of halophilic *P. aeruginosa* EtOAc crude extract (HPAEtOAcE) was detected with a progressive increase in the inhibition zone against MRSA, from 80 to 140 µL, with inhibition zone diameters of 15 and 21 mm, respectively (Figure 1a). The maximum was found to be 21 mm, whereas the minimum was 9.0 mm. The highest inhibition zone showed elevated antibacterial potential of the extract. The MIC of HPAEtOAcE was found to be 0.64 µg/mL and MBC was found to be 1.28 µg/mL, noted with complete inhibition of bacterial growth and no visible growth. This indicated that the active extract had strong antibacterial potential against MRSA.

### 3.2. Microbial Biomass Inhibition Assay and Bacterial Cell Live/Dead Assay 

The bioactive efficacy of HPAEtOAcE on the microbial biomass of MRSA is shown in Figure 1b. Results indicated that the microbial biomass was greatly reduced at the higher concentration of 150 µg/mL of extract, which effectively killed bacterial cells in suspension with 0 OD value, whereas the lower concentration of 75 µg/mL also significantly killed bacterial cells, with 0.491 ± 0.001 OD_660_. The 100% inhibition percentage was effectively attained at a high concentration, which denotes the potential antimicrobial efficiency of the extract and 72.92% inhibition was observed at low concentrations. In the live/dead assay, the highest number of dead cells were observed at 150 µg/mL, emitting red fluorescence in epi-fluorescence microscopy by observing PI; significant quantities of dead cells were also observed at 75 µg/mL (Figure 2a).

### 3.3. Scanning Electron Microscopy 

Significant morphological changes were observed by SEM analysis at a concentration of 150 µg/mL of the HPAEtOAc crude extract in the mid-exponential growth phase of the MRSA (Figure 2b). Shrinking of the cell shape and membrane disruption was also observed in the cells. In contrast, no morphological changes were noted in the control and the remaining cell shape was a coccus. The high concentration treatment of cells revealed the devastating effects on the cell membrane, the reduced shape of the cells, and the significant cleavage at the cell surface. This represents the activity of HPAEtOAcE potentially killing the bacteria by damaging the cell membrane at higher concentrations (150 µg/mL). A lower concentration (75 µg/mL) of the cells also caused considerable cell damage.

### 3.4. Biofilm Inhibition Assay—Confocal Laser Scanning Microscopy (CLSM)

The biofilm inhibition efficacy of active HPAEtOAcE was demonstrated in MRSA cells treated at 150 µg/mL at 48 h, resulting in no biofilm formation (OD_600_ 0.002 ± 0.001) compared with the control and before treating cells at various intervals of 6–24 h, which provided a gradual decrease in biofilm formation at the respective OD. Similarly, the 3D images revealed a clearer picture of active HPAEtOAcE biofilm inhibition (Figure 3). The SYTO9 dye was observed through the produced biofilms of cells and displayed a green fluorescence; under CLSM microscopic observation, it also exhibited the adhesive and non-adhesive effects of MRSA cells due to the presence of potential extracts (Figure 4a)

### 3.5. Hemolytic Activity

For the in vitro hemolytic action on human erythrocytes, different amounts of HPAEtOAcE were utilized. Complete hemolysis was obtained in the H_2_O_2_ as a positive control with 100% hemolysis (OD_540_ 0.694 ± 0.002), whereas varied concentrations of HPAEtOAcE showed no hemolysis on erythrocytes (Figure 4b). Significantly greater concentrations of HPAEtOAcE (20 mg/mL dissolved in milli-Q water) generated much less hemolysis (0.865%) as none of the minimal concentrations possessed hemolysis, except for 10 mg/mL (0.102%) The extract revealed very little hemolytic levels on human erythrocytes, only at 20 mg/mL (OD_540_ 0.002 ± 0.001). These results demonstrated that HPAEtOAcE did not have significant detrimental effects on human red blood cells, and that, even at high concentrations, it showed very little hemolytic activity.

### 3.6. Antioxidant Assay

Compared with ascorbic acid (a standard antioxidant compound), HPAEtOAcE revealed substantial antioxidant activity. The IC_50_ of HPAEtOAcE was obtained as 29.75 ± 0.38, which is slightly increased compared with ascorbic acid IC_50_, determined as 25.21 ± 0.35 (Figure 4c). DPPH scavenging activity of HPAEtOAcE was 88.618 ± 1.63% at 60 mg/mL and ascorbic acid was 90.244 ± 1.41%. This showed that the HPAEtOAcE had effective antioxidant activity, and therefore, could be a potential natural agent that could be applied in pharmaceutical applications similar to ascorbic acid.

### 3.7. Qrt (Real Time)—PCR 

The effect of HPAEtOAcE against the targeted virulence genes of MRSA was examined as per the respective role of genes provided by Ma et al. (2012); the primers used for the selective genes are given in Table 1. *EbpS* was observed to be slightly upregulated among the four selected genes, whereas *hlA*, *IrgA*, and *SpA* were dramatically downregulated (Figure 5). The RQ value vs. targeted genes clearly revealed that gene regulation gradually declined toward negative values and no significant differences were observed in the control. Thus, the potent in vitro antimicrobial effects of the EtOAc extracts could be attributed to one of the major processes of downregulating selected virulence genes.

### 3.8. Identification of Compound

Based on the literature and characterization analysis, the hexane HPAEtOAcE fraction of *Pseudomonas aeruginosa* contained a new compound, which was identified as 5-(1*H*-indol-3-yl)-4-pentyl-1,3-oxazole-2-carboxylic acid (Figure 6). The novelty of this compound was confirmed by SciFinder. Its HPLC, LC-MS, and ^1^H, ^13^C NMR characterizations are shown in the Appendix A).

### 3.9. Characterization of 5-(1h-Indol-3-yl)-4-Pentyl-1,3-Oxazole-2-Carboxylic Acid (Compound ***1***)

Results of ^1^H NMR (400 MHz, DMSO-d6, ppm): δ 0.88 (3H, t, J = 6.5 Hz, -CH3 group), 1.28–1.89 (4H, m, -CH2, -CH2), 2.94 (2H, t, J = 7.4 Hz, -CH2), 7.31- 7.94 (4H, m benzene molecule), 8.51 (1H, d, pyrrole), 12.36 (1H, s, -NH), and 12.63 (1H, s, OH). Results of ^13^C NMR (100 MHz DMSO-d6): δ 169.2 (COOH), 162.5 (1,3-oxazole), 155.6, (1,3-oxazole), 137.26 (pyrrole), 136.1 (pyrrole), 124.4 (1,3-oxazole), 122.7–106.3 (benzene), 34.35 (-CH2), 28.7 (-CH2), 28.39 (-CH2), 22.6 (-CH2), and 14.0 (-CH3) (Appendix A). MS (ESI) m/z (% of relative abundance) calculated for [C17H18N2O3] + 298.34 was 299.51 [M+H]. Results of IR (KBr) in cm^−1^: 3286 (carboxylic acid, OH stretch), 2983 (C-H stretching, alkane), and 1660 (C=C stretching). The purity of the isolated compound was confirmed by HPLC at a retention time of 2.61. The confirmed chemical structure of the isolated compound 5-(1h-indol-3-yl)-4-pentyl-1,3-oxazole-2-carboxylic acid (Compound **1**) is shown in Figure 6.

### 3.10. Binding Affinity of Compound ***1*** through Molecular Docking 

A molecular docking study was carried out to investigate the molecular interaction pattern of the isolated Compound **1** with the pantothenate synthetase protein of MRSA (PDB ID: 2 × 3F). From the results of the docking study, Compound **1** showed good binding affinity at the active site of pantothenate synthetase and the CDOCKER energy was −32.34 kcal mol^−1^. This higher interaction can be explained by the formation of three H-bonds, five alkyl, and one Pi-sulfur interaction with respective amino acids in pantothenate synthetase (Figure 7). In particular, the main functional COOH group of Compound **1** formed two H-bonds with Arg 188 and Lys 150, respectively. In addition, the aliphatic and aromatic groups in Compound **1** formed van der Waals interactions with most of the active sites of the amino acids. The docking results confirmed that the isolated new Compound **1** effectively interacted with the pantothenate synthetase protein of MRSA.

### 3.11. Molecular Dynamic Simulations

The optimal analysis of (5-(1h-indol-3-yl)-4-pentyl-1,3-oxazole-2-carboxylic acid with 2 × 3F protein for molecular dynamics simulation was examined based on the docking results. The simulation was done on binding complex systems to investigate the dynamic behavior of the targeted protein. During the solvation method, NaCl and water molecules were introduced to the protein to prepare it for further stability testing (Figure 8a). Quality-check parameters for the simulated system (temperature, total energy, RMSD, and RMSF) were analyzed to ensure simulation validity. Figure 8b illustrates temperature changes and total energy calculations of the protein complex in MD simulation analysis. During the 300 ps MD simulation period, the systems were stabilized and showed no significant changes in temperature or the total energy analysis in between the unbound bound of the 5-(1h-indol-3-yl)-4-pentyl-1,3-oxazole-2-carboxylic acid with protein. Finally, the temperature and stability analysis of the 2 × 3F complex in MD analysis indicated that the 2 × 3F with (5-(1h-indol-3-yl)-4-pentyl-1,3-oxazole-2-carboxylic acid complex was more stable.

In RMSD analysis (Figure 8c), the unbound protein 2 × 3F and the bound (5-(1h-indol-3-yl)-4-pentyl-1,3-oxazole-2-carboxylic acid with 2 × 3F showed no significant variations in RMSD. These results confirmed that the isolated compound did not induce any major changes in the 2 × 3F protein active site. Further, the RMSF investigated the influence of (5-(1h-indol-3-yl)-4-pentyl-1,3-oxazole-2-carboxylic binding on the flexible region of the targeted protein 2 × 3F. The RMSF results showed that the active sites of amino acids (from 30 to 200 regions) in the 2 × 3F complex did not show any fluctuations when compared with the unbound protein 2 × 3F (Figure 8c). In conclusion, the docked (5-(1h-indol-3-yl)-4-pentyl-1,3-oxazole-2-carboxylic acid molecule at the 2 × 3F showed good binding affinity and did not change the stability, total energy, RMSD, or RMSF of the target 2 × 3F protein.

### 3.12. ADMET and Toxicity Prediction

The identified Compound **1** was submitted to ADMET and TOPKAT protocol to assess the pharmacokinetics (hepatotoxicity levels, aqueous solubility, cytochrome CYP2D6 inhibition, plasma protein binding (PPB), blood-brain-barrier penetration (BBB), human intestinal absorption (HIA), and toxicity) and drug-likeness (Lipinski rule of five) properties. From the results of the pharmacokinetic analysis (Figure 9), Compound **1** had low BBB penetration, good absorption and solubility, and PPB (<90%) in nature. From the TOPKAT results (Table 2), Compound **1** did not contain any toxic fragments nor did it show a positive response to carcinogenicity nor other toxicities. Furthermore, Compound **1** followed the Lipinski rule of five (molecular mass: 500 Dalton; high lipophilicity: 0.19; H-bond donor 0; and H-bond acceptor: 5), indicating it had better drug-likeness properties.

## 4. Discussion

Much of the genetic diversity of MRSA occurs within the accessory genome, where common mediators of virulence, immune evasion, and antibiotic resistance are found. MGEs carrying antibiotic-resistance genes have been acquired by MRSA on multiple independent occasions [42]. The high incidence of MRSA has recently brought greater attention on hospital-acquired infections [2]. In the current investigation, no zone formation in the methicillin strip supported the resistance patterns of hospital-derived *S. aureus*. However, there is an urgent need to control the infections associated with MRSA in hospitals as well as in the community, which necessitates the use of effective antibiotics against MRSA. Anti-MRSA compounds were discovered from a variety of sources, including marine environments ([43,44,45]. The halophilic niche has recently sparked interest due to its potential in providing compounds against drug-resistant microorganisms. [28,31]). Our previous findings described the isolation of halophilic *Pseudomonas aeruginosa* from a saltpan and the extraction of bioactive compounds using EtOAc against drug-resistant bacteria isolated from diabetic foot patients [31]. In the current study, the same extract was tested against MRSA, and the MIC and MBC values were slightly supportive of findings from Tharmalingam et al. [46]. Furthermore, MBCs were obtained close to their identified compounds: 1 and 4 were 64 and 32 g/mL, respectively, and MICs were 2 and 4, respectively. Our results showed small increases in MIC at 64 and MBC at 1.28 µg/mL. Pereira et al. [47] found that halophilic bacterial strains had higher antimicrobial responses against Gram-negative bacteria than Gram-positive, which was consistent with findings of Pelaez et al. that found that Gram-positive strains were more susceptible to antibiotics than Gram-negative strains [48].

Our previous findings [31] showed that HPAEtOAcE crude extract was found to be higher in killing gram-negative strains such as *K. quasivariicola* (64 µg/mL) and *E. coli* (32 µg/mL) than *S. argenteus* (16 µg/mL). MRSA virulence, on the other hand, competes with antibiotics with its complete force to exhibit antibiotic resistance. [42]. The microbial biomass inhibition results showed the devastating killing efficiency of HPAEtOAcE crude extract on MRSA at 150 µg/mL and 0 OD_600_ at 100% inhibition, and the lowest at 72.92% inhibition. Al-Dhabi et al. observed that the EtOAc extract of marine *Streptomyces* sp showed 100% killing of Gram-negative strains, such as *P*. *aeruginosa*, *E. coli*, and *K. pneumoniae*, and approximately 80% killing of Gram-positive strains, such as *S. epidermidis* and *S. aureus*, at 100 µg/mL [37]. The current investigation demonstrated 100% biomass inhibition of MRSA. The bacterial live and dead assay efficiently assessed the total killing effectiveness of active HPAEtOAcE at concentrations of 150 g/mL. The destruction of the cell envelope by antibiotics stressed the cells to death, and the dual staining of PI and SYTO9 revealed the dead and alive stages of MRSA cells. The SYTO9 membrane permeable dye is mostly used in live and dead bacterial biofilm observations. It selects total bacterial populations, whereas the PI selects only intact cells with conciliated membranes [49]. The green fluorescence emission in control cells represents live cells that had not been treated with antimicrobials; the highest concentration of 150 µg/mL revealed dead cells, whereas the lower concentration of 75 µg/mL determined the PI permeability toward disturbed membranes to be nonviable cells [39]. The results suggested that the HPAEtOAcE penetrated bacterial cell membranes and caused cell death, indicating that it could be a mode for PI penetration when dead cells are considered. Similarly, biofilm inhibition was achieved at the 150 µg/mL concentration, which resulted in complete inhibition under confocal microscopic observation at 48 h incubation. However, different interval times from 6–48 h showed a gradual decrease in biofilm formation [39,50]. The dye SYTO9 was used to observe the cells, as well as the developed biofilms in control and treated cells, with maximum and reduced effects of biofilms on MRSA. The biofilm-inhibiting effects on MRSA by various bioactive compounds has already been shown in several in vitro studies [50].

The susceptibility of cells and biofilm matrices to antimicrobial agents is age-dependent; Vidakovic et al. [51] stated that biofilm age is a highly influential factor for antimicrobial susceptibility. Younger biofilms (24 h) are more susceptible to antimicrobial agents than older biofilms (48 h) [52]. At 150 µg/mL, the HPAEtOAcE effectively inhibited both the 24 and 48 h biofilms. The biofilm matrix inhibition effects were also found to be sustained for 48 h, beginning at 6 h of incubation after adding HPAEtOAcE. This effect indicated that the active extract was more effective on biofilm inhibition, even after 48 h of biofilm formation which is highly supportive of findings from Warraich et al., who found dispersal effects of biofilms using various amino acid additives [50]. According to Samrot et al., various compounds including phenolic acids, flavonoids, lectins, tannins, and alkaloids are actively involved in biofilm inhibition by inhibiting QS (quorum sensing) molecules on both Gram-positive and Gram-negative bacteria [53]. The HPAEtOAcE, containing biochemical compounds, was reported in our previous study in the GC/MS profile [31], and impacted the present work on MRSA biofilms due to the presence of bioactive compounds mentioned above that affected MRSA [54,55]. The lowest concentration of 62.5 µg/mL of the crude EtOAc extract of marine *Streptomyces* sp. SBT348 inhibited biofilm formation by 90% [56]. Our study used the higher concentration of 150 µg/mL of HPAEtOAcE, which effectively inhibited the biofilm at 99.89% at 48 h; this supports findings from Shinde et al. who reported that epigallocatechin-3-gallate-stearate, combined with erythromycin and tetracycline antibiotics for a synergistic effect, caused 95–99% of biofilm inhibition on Gram-positive and Gram-negative bacteria [39]. The current study showed a similar effect with HPAEtOAcE alone on MRSA without adding any antibiotics. Manilal et al. found that an ethanolic extract of 1000 µg/mL of the *Moringa stenopetala* plant completely prevented a MRSA biofilm, which had a higher concentration than in the current investigation [57].

According to SEM results, MRSA in TSB medium as a control showed round undamaged cells, and after incubation with a minimum and maximum concentration of HPAEtOAcE, the bacterial cells appeared to shrink, with multiple dents in their cell wall, and many cells were lysed at the higher concentration compared with the lower concentration. Some deep craters were also observed in the cells, indicating that the cells had burst in the treated cells, but more so at high concentrations. In the figures, cell debris was also observed alongside cell lysis [54]. 

In the present study, anti-hemolytic activity was observed at all concentrations with 0 OD value exceptions of 20 mg/mL, which showed 0.28% hemolysis; it showed 0% hemolysis activity at other concentrations. It was clearly demonstrated that the active HPAEtOAcE had no toxic effect on human erythrocytes in a dose-dependent manner [58]. As all of the bioassay studies were carried out at low concentrations, the concentrations employed in the hemolytic activity tests were greater. The present study found that all concentrations, except 20 mg/mL, were protective against hemolysis on the erythrocyte membrane in a dose-dependent manner [59]; the protection of RBCs leads to a delay in hemolysis [60,61]. Our findings were slightly supportive of Chi et al., who found that the marine-derived fungal EtOAc extract provided pronounced hemolytic activity at concentrations ranging from 20 to 200 µg/mL, but our results showed significantly less hemolytic activity than their concentrations [62].

In terms of antioxidant properties, the HPAEtOAcE expressed potential antioxidant responses, measured by the DPPH assay, and the IC_50_ was consequently determined to be 29.75 ± 0.38 mg/mL. Potential biochemicals, including phenolic compounds and acid derivatives that were represented in the GC/MS profile of our previous report, could be a responsible factor for this activity [31]. Marine organisms in extreme conditions, including high salinity, temperature, and pressure, produce metabolites for their survival and other proliferation activities. The pigments from halophilic bacteria providing anticancer and antioxidant properties most frequently are bacterioruberin [28,63,64,65,66,67,68,69,70]. As a result, the present study revealed the antioxidant potential of HPAEtOAcE as 88.618 ± 1.63% of DPPH scavenging, which is more supportive with the marine *Streptomyces variabilis*-derived EtOAc extract, given its 82.86% of DPPH scavenging at 5 mg/mL, which is much lower than our concentration (60 mg/mL) [71].

The gene expression analysis of RT-PCR revealed that the expression of the *hlA* gene responsible for toxin synthesis in MRSA was downregulated toward the RQ (real-time quantification) of 0 to −5. Similarly, the *IrgA* (murein hydrolase regulator) and *SpA* (binding protein immunoglobulin G) genes, which are responsible for cell death and lysis, and protein secretion for bacterial aggregation were also downregulated from 0 to −2.5 compared with the control (*p* < 0.05). The virulence genes were chosen based on the findings of Ma et al. (2012), who found downregulation of virulence genes, indicated by a decrease in gene expression on various *S. aureus* genes, such as sar, icaD, icaA, and bap [72,73]. The sub MBIC (minimum bactericidal inhibition concentration) of 0.098 mg/mL ethanolic extracts of *M. communis* decreased the level of expression in the aforementioned genes, ranging from two- to three-fold. In the current study, 150 µg/mL of HPAEtOAcE effectively downregulated the targeted genes, including *hlA*, *IrgA*, and *SpA*, also at two- to three-fold decreases as in previous findings. These results revealed that the quantities of the responsible genes for the aforementioned cellular components were reduced. In contrast, the *Ebps* gene, which is responsible for surface protein synthesis, was upregulated to a positive value of 0–2.5. Findings from Ma et al. were consistent with the present findings of *SpA* gene downregulation [74]. According to their report, the compound CCG-203592 decreased *SpA* gene expression at the mid-logarithmic, late logarithmic, and stationary phases; the *hlA* and *IrgA* genes were also reported to be decreased, but no significant changes were seen in the *Ebps* gene. 

Furthermore, the HPAEtOAcE, containing novel carboxylic acid, was confirmed via chromatographic and NMR analyses [75]. The functional group of carboxylic acid has importance for drug design because nearly >450 marketed drugs include molecules with carboxylic acid; the isosteres of carboxylic acid were also analyzed to better understand their physical properties [76,77]. From the year 2000 to the present, the antimicrobial properties of various carboxylic acids have been thoroughly researched. Some examples include: *cis*/*trans*-3-aryl(heteroaryl)-3,4-dihydroisocoumarin-4-carboxylic acids (**3a–i**) D, which showed antibacterial properties against *S. aureus* and *A. niger* [78]; carboxylic acid showed toxic effects on bacterial growth [79]; *Lactobacillus* produced 2-pyrrolidone-5-carboxylic acid and exhibited a wide growth inhibition on *P. putida* 1560-2, *Pseudomonas fluorescens* KJL G, and *Enterobacter cloacae* 1575 [80]; and the fungus *Aspergillus fumigatus* nHF01 produced 5-butyl-2-pyridine carboxylic acid, revealing a broad spectrum of antimicrobial activity against tropical and food pathogenic bacteria [75]. Antimicrobial activity of N-alkylimidazole-2-carboxylic acid derivatives against *E. coli, S. aureus,* and *Candida albicans* was also demonstrated [81]. Similarly, several carboxylic acid derivatives and salt additives, including sodium and potassium, have a broad spectrum of antimicrobial properties and inhibition of key enzymes [82,83,84,85,86]. The present study also demonstrated that the novel halo-derived carboxylic acid showed antibacterial activity against MRSA. A molecular docking study of our novel compound with pantothenate synthetase on MRSA slightly correlated with the findings of Samala et al. [87]. These authors studied the binding interactions of imidazo [2,1-b] thiazole and ben-zo[d] imidazo [2,1-b] thiazole derivatives as effective inhibitors on the active site of pantothenate synthetase. More specifically, the hydrogen bond interaction with amino acids, such as Ser196, Ser197, Gln72, HIE44, HIE47, and Asp161, had excellent binding energy between −7.15 to −8.50 kcal/mol and showed stability with the enzyme pantothenate synthetase of *Mycobacterium tuberculosis*. This was correlated with the present investigation due to the hydrogen bond interactions, but the binding energy was found to be higher in our results (−32.34 kcal mol^−1^). This higher interaction can be explained by the formation of three H-bonds, five alkyl, and one Pi-sulfur interaction with respective amino acids in PS.

Quercetin derivatives were also shown to excellently bind with pantothenate synthetase of *M. tuberculosis* [88] and its derivatives, including (2 S)-5,7-dihydroxy-2-{3-hydroxy-4-[(4-methylphenyl) sulfonyl] phenyl}-2 H-chromene-3,4-dione and 3-amino-4-{5-[(2)-5,7-dihydroxy-3,4-dioxo-3,4-dihydro-2H-chromen-2-yl]-2,3-dihydroxyphenyl}-4-oxobutanamide, which produced total binding energies of −131.572 and −123.464 kcal/mol, respectively, with H-bond energy of −18.7922 and −23.1897 kcal/mol, respectively. The sulfamoyl adenylate inhibitors also showed docking ability to the enzyme pantothenate synthetase of *M. tuberculosis* with a glide score of −13.014601 to −8.516113 [89]. In the MD analysis of our results, the docked (5-(1h-indol-3-yl)-4-pentyl-1,3-oxazole-2-carboxylic acid molecule in 2 × 3F showed good binding affinity and did not change the stability, total energy, RMSD, and RMSF of the target 2 × 3F protein. It was significantly stronger than the values found by Masumi et al. [90], who found that the docked system of Kaempferol 3-rutinoside-7-sophoroside and rutin compounds provided more stability for 10 ns MD simulations with the MRSA penicillin-binding protein 2a, whereas our docked system exhibited 300 ps MD simulations with the MRSA PS protein. The in silico studies showed the MRSA inhibitory activity of Compound **1,** which was also shown in the in vitro study. The identified Compound **1** showed better drug-likeness properties following the Lipinski rule of five (molecular mass: 500 Dalton, high lipophilicity: 0.19, H-bond donor: 0, and H-bond acceptor: 5), that was consistent with the results of Omar et al. [91]. Additionally, there was no hepatotoxicity and the compound did not inhibit the enzyme cytochrome CYP2D6. The MD simulation test revealed that the stability of the protein with Compound **1** had not been altered. [92]. The ADMET and TOPKAT protocol accessed the pharmacokinetics and drug-likeness (Lipinski rule of five) properties of Compound **1** with nontoxic effects [93,94]. However, beneficial metabolites have been discovered, albeit rarely, from a few *Pseudomonas* species, such as an anti-MRSA compound from marine *P. aeruginosa* UJ-6, compounds against plant pathogens [34,35,95,96], and a recently discovered compound with bactericidal effects and biofouling control [97]. The current study could be a beneficial database for the halophilic *P. aeruginosa*, which secretes a bipotent novel antibacterial substance against MRSA and may be a promising antimicrobial agent for future research. In addition, the in vivo study will also be applicable in the future and the compound may be applied to clinical trials based on the results. The novel carboxylic acid has also been characterized for its non-hemolytic and antioxidant potential in the current study, and this could be applicable in the future as a promising non-toxic drug. The in silico toxicity prediction confirmed this compound’s drug-likeness properties, indicating its potential as an enhancing drug against MRSA infections in the future.

## 5. Conclusions

Antibiotic resistance has increased in hospitals because of increased antibiotic consumption. However, human populations are associated with chronic illness when it comes to community effects due to antibiotic overuse. Antibiotic treatment failure in human populations results in the emergence of drug resistance among HAI. MRSA infection is a serious threat to humans and necessitates the use of more effective antibiotics with a higher killing efficiency to suppress its many virulence factors. Solar salterns are emerging as important sites for the extraction of bioactive compounds in the search for novel metabolites. The current study demonstrated the value of a novel carboxylic acid, extracted from halophilic *P. aeruginosa* isolated from solar salterns, that demonstrated broad antimicrobial activity against MRSA in both in vitro and in silico analyses ranging from antimicrobial activity to molecular docking, with excellent binding affinity for the MRSA-PS protein. Despite the significant virulence associated with *P. aeruginosa* in the clinical sector, only a few reports have recently emerged due to the species synthesizing some beneficial bioactive compounds. Carboxylic acid is commonly found in most fruits and vegetables as a highly eco-friendly compound associated with human populations. The novel finding of the halophilic *P*. *aeruginosa*-derived carboxylic acid was investigated using HPLC, LC/MS, and NMR study, and showed great bioactivity including antibacterial effects, MRSA biomass inhibition at 24 h (100%), biofilm inhibition at 48 h (99.89%), downregulation of virulence genes, anti-hemolytic effects, antioxidant effects, and excellent binding affinity with MRSA-PS protein amino acids. The compound stability with the PS protein was confirmed by RMSD value, and the drug-likeness and ADMET toxicity prediction revealed that the compound is non-toxic. These findings indicate that the novel 5-(1H-indol-3-yl)-4-pentyl-1,3-oxazole-2-carboxylic acid (Compound **1**) could be used as an effective antibiotic against MRSA as a safe ligand. Furthermore, it could be used in the future as an effective drug with high orientation against all pathogenic microbes. More in-depth studies are warranted in the future to propagate the novel compound as a promotable drug against several harmful pathogens.

## Figures and Tables

**Figure 1 metabolites-12-01094-f001:**
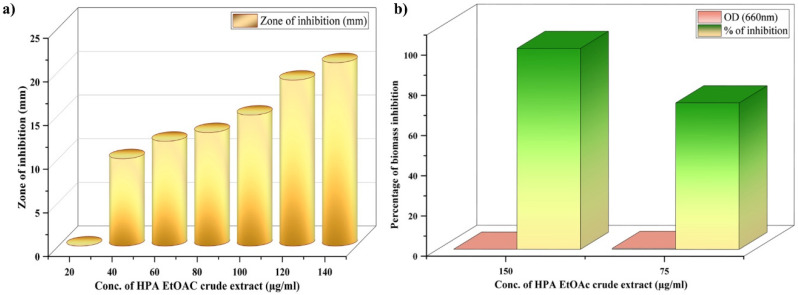
(**a**) Antibacterial activity of halophilic *P. aeruginosa* derived EtOAc (HPA EtOAc) crude extract in zone of inhibition (mm in diameter) against methicillin-resistant *Staphylococcus aureus* (MRSA); (**b**) microbial biomass inhibition efficiency of HPA EtOAc crude extract at 75 and 150 µg/mL concentrations on MRSA.

**Figure 2 metabolites-12-01094-f002:**
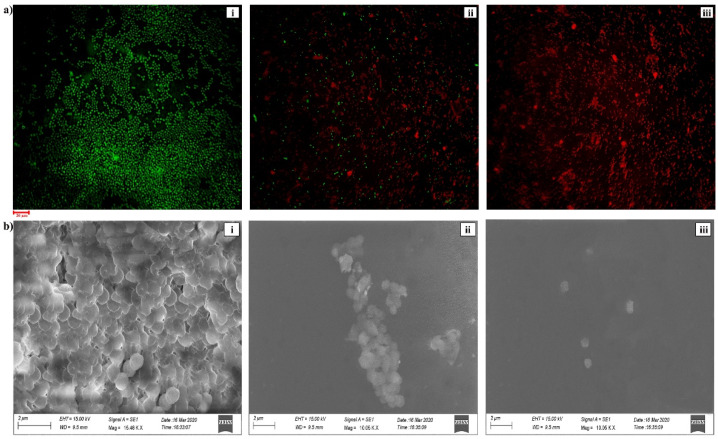
(**a**) CLSM image of bacterial live/dead assay on MRSA provided by HPAEtOAc crude extract: (i) control, (ii) treated at 75 µg/mL, and (iii) treated at 150 µg/mL concentration on MRSA. (**b**) SEM image of bacterial alive/dead assay on MRSA provided by HPAEtOAc crude extract: (i) control, (ii) treated by 75 µg/mL, and (iii) treated by 150 µg/mL concentration on MRSA.

**Figure 3 metabolites-12-01094-f003:**
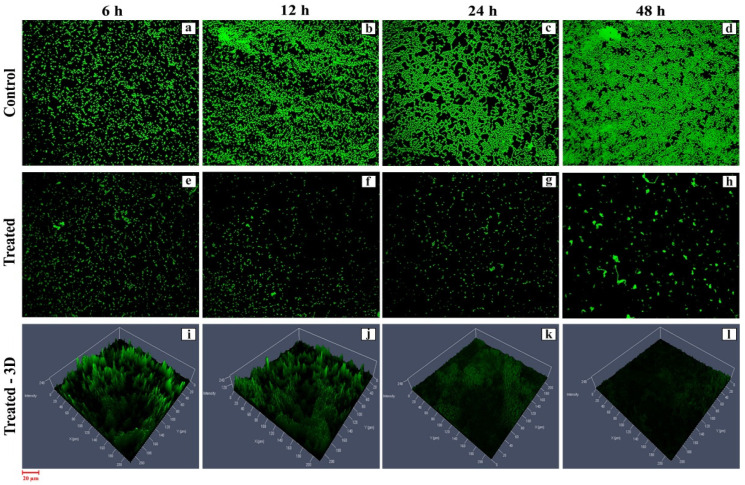
CLSM image of biofilm inhibition efficiency provided by the HPAEtOAc crude extract on MRSA cells at 6, 12, 14, and 48 h: (**a**–**d**) control; (**e**–**h**) treated; (**i**–**l**) treated 3D view.

**Figure 4 metabolites-12-01094-f004:**
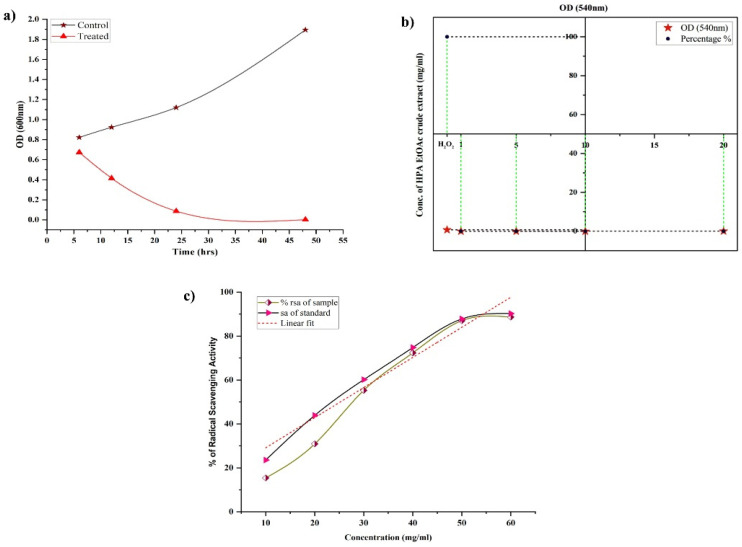
(**a**) Biofilm inhibition efficiency provided by HPAEtOAc crude extract on MRSA cells at 6, 12, 14, and 48 h at 570 nm OD between the control and treated MRSA cells at 150 µg/mL. (**b**) Anti-hemolytic activity of HPAEtOAc crude extracts on human erythrocytes up to 1–10 mg/mL, showing a non-hemolytic response. (**c**) Antioxidant activity of HPAEtOAc crude extracts using a DPPH free radicals scavenging assay, expressed as radical scavenging activity (RSA) vs. concentration of HPAEtOAc crude extract.

**Figure 5 metabolites-12-01094-f005:**
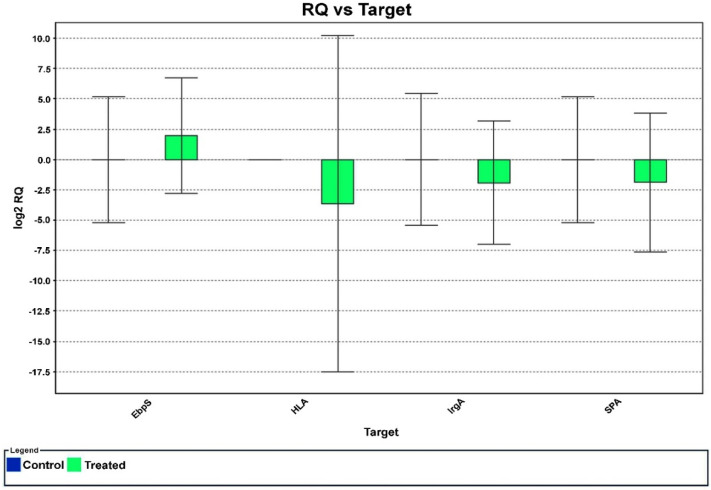
The effect of HPAEtOAc crude extract on selected MRSA gene expression in real−time PCR. The negative values denote the downregulation of targeted genes at 150 µg/mL of HPAEtOAc crude extract.

**Figure 6 metabolites-12-01094-f006:**
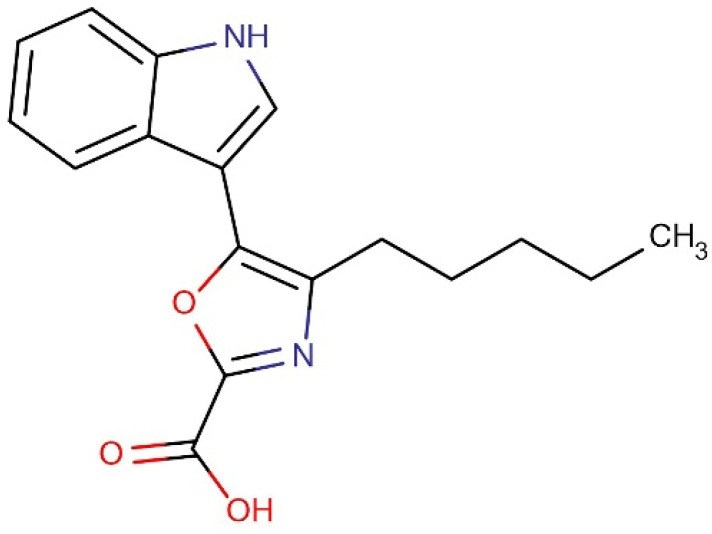
Structure of the ethyl acetate fraction containing the novel compound 5-(1*H*-indol-3-yl)-4-pentyl-1,3-oxazole-2-carboxylic acid.

**Figure 7 metabolites-12-01094-f007:**
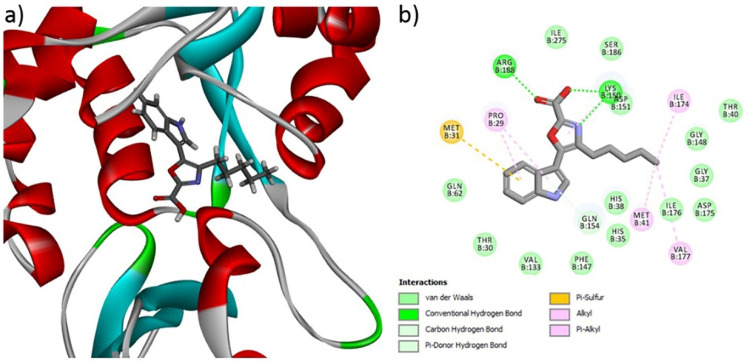
(**a**) 3D and (**b**) 2D interactions of Compound **1** at the active site of the pantothenate synthetase protein of MRSA.

**Figure 8 metabolites-12-01094-f008:**
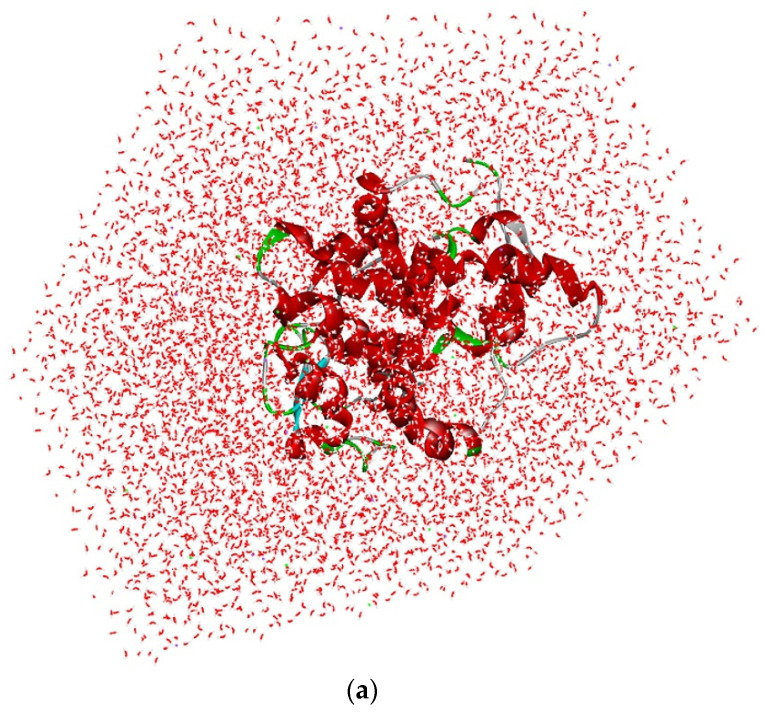
(**a**) Solvation analysis of 2 × 3F protein with (5-(1h-indol-3-yl)-4-pentyl-1,3-oxazole-2-carboxylic acid. (**b**) Temperature and total energy analysis of 2 × 3F protein with (5-(1h-indol-3-yl)-4-pentyl-1,3-oxazole-2-carboxylic acid. (**c**) RMSD and RSMF analysis of 2 × 3F protein with (5-(1h-indol-3-yl)-4-pentyl-1,3-oxazole-2-carboxylic acid.

**Figure 9 metabolites-12-01094-f009:**
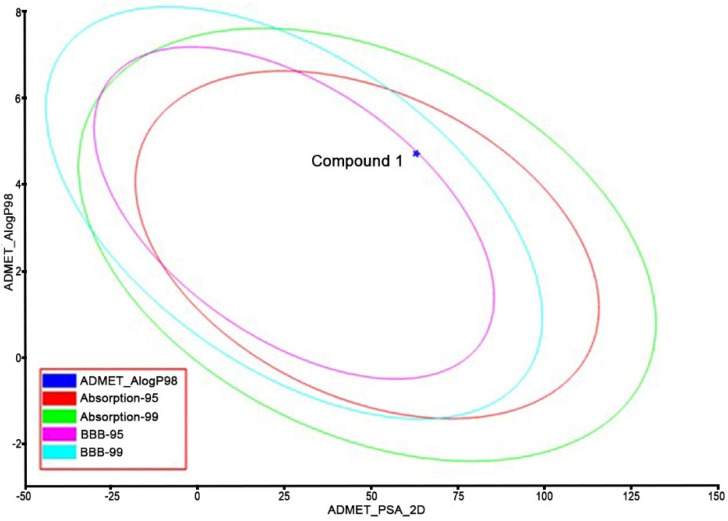
Pharmacokinetic analysis of Compound **1** (TOPKAT).

**Table 1 metabolites-12-01094-t001:** Primers used for selective gene factors responsible for MRSA virulence, tested by RT-PCR.

S. No	Primers	Gene
1	F: CTGGTAGTCCACGCCGTAAAC R: CAGGCGGAGTGCTTAATGC	16S rRNA standard gene
2	F: GCGCAACACGATGAAGCTCAACAA R: ACGTTAGCACTTTGGCTTGGATCA	SPA
3	F: CTGGTGCTGTTAAGTTAGGCGAAG R: GGCTGGTACGAAGAGTAAGCCAAT	IrgA
4	F: TTTCCGGTGAACCTGAACCGTAGT R: ACAGCAACAACAACGTCAAGGTGG	ebpS
5	F: CTGAAGGCCAGGCTAAACCACTTT R: GAACGAAAGGTACCATTGCTGGTCA	Hla

**Table 2 metabolites-12-01094-t002:** Pharmacokinetic analysis for Compound **1** (TOPKAT).

S. No	Property	Prediction
1.	Aerobic Biodegradability	Non-Degradable
2.	Ames Mutagenicity	Non-Mutagen
3.	Developmental Toxicity Potential	Non-Toxic
4.	Mouse Female FDA	Non-Carcinogen
5.	Mouse Female NTP	Non-Carcinogen
6.	Mouse Male FDA	Non-Carcinogen
7.	Ocular Irritancy	Non-Irritant
8.	Rat Female FDA	Non-Carcinogen
9.	Rat Female NTP	Non-Carcinogen
10.	Rat Male FDA	Carcinogen
11.	Rat Male NTP	Non-Carcinogen
12.	Skin Irritancy	Non-Irritant
13.	Skin Sensitization	Non-Sensitizer
14.	Weight of Evidence Rodent Carcinogenicity	Non-Carcinogen
15.	Carcinogenic Potency TD_50_ Mouse	42.9
16.	Carcinogenic Potency TD_50_ Rat	53
17.	Chronic LOAEL	0.666
18.	Daphnia EC_50_	5.34
19.	Fathead Minnow LC_50_	0.00141
20.	Rat Inhalational LC_50_	3.85
21.	Rat Maximum Tolerated Dose Feed	0.712
22.	Rat Maximum Tolerated Dose Gavage	0.246
23.	Rat Oral LD_50_	5.46

## Data Availability

The data presented in this study are available in article.

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
