# Peer review of "Bioactive Efficacy of Novel Carboxylic Acid from Halophilic *Pseudomonas aeruginosa* against Methicillin-Resistant *Staphylococcus aureus"

_metabolites, 2022, doi:10.3390/metabo12111094_

Round 1
Reviewer 1 Report
The manuscript is interesting and the current topic is related to the search for new antibacterial substances that are not antibiotics. Nevertheless, I have a few comments:
- I strongly suggest you to proofread English; in some places there are incorrect or colloquial wording and grammatical errors
Abstract: the conclusion is poorly expressed
Saltpan bacteria (lines from 123): a little more information should be introduced regarding the extract used. The authors referred only to the previous article, but it is difficult to assess the properties of the substance used.
Line 328: It is unclear how the authors measured the inhibition zone. The diameter of the well was 6mm, so where did the 0.9mm zone come from?
Line 550: How can virulence be controlled? The authors' actions relate rather to drug resistance.
Minor proofreading notes:
line 326: please fill in µL
line 366: iii- please change the description order
Fig 5 is not clear
Reviewer 2 Report
I have enjoyed reading manuscript entitled “Bioactive efficacy of Novel Carboxylic acid from Halophilic Pseudomonas aeruginosa against Methicillin resistant Staphylo- 3 coccus aureus.” The problem of AMR is amplifying days by day in developed and developing countries. Discovery of new antibiotics is one of the ways to help cope this alarming situation. The current manuscript has addressed the problem of AMR.
The manuscript has provided all integral components in the methods section. I have only a few comments that could help further polish this manuscript.
I would suggest that the authors add precise information about the guidelines with a modified protocol of Buzgaia et al.
Is it possible for the authors to discuss in detail the future roadmap regarding commercialization of this product? Are you planning for preclinical studies and clinical trials for this new compound? If yes, what could be the success rate in this regard?
Round 2
Reviewer 1 Report
I have no any other comments
Reviewer 2 Report
No further comments